# Role of Genetic Variation in Transcriptional Regulatory Elements in Heart Rhythm

**DOI:** 10.3390/cells13010004

**Published:** 2023-12-19

**Authors:** Timo Jonker, Phil Barnett, Gerard J. J. Boink, Vincent M. Christoffels

**Affiliations:** 1Department of Medical Biology, Amsterdam Cardiovascular Sciences, Amsterdam University Medical Centers, 1105 AZ Amsterdam, The Netherlands; t.jonker@amsterdamumc.nl (T.J.); p.barnett@amsterdamumc.nl (P.B.); g.j.boink@amsterdamumc.nl (G.J.J.B.); 2Department of Cardiology, Amsterdam Cardiovascular Sciences, Amsterdam University Medical Centers, 1105 AZ Amsterdam, The Netherlands

**Keywords:** regulatory elements, GWAS, arrhythmia, SNPs

## Abstract

Genetic predisposition to cardiac arrhythmias has been a field of intense investigation. Research initially focused on rare hereditary arrhythmias, but over the last two decades, the role of genetic variation (single nucleotide polymorphisms) in heart rate, rhythm, and arrhythmias has been taken into consideration as well. In particular, genome-wide association studies have identified hundreds of genomic loci associated with quantitative electrocardiographic traits, atrial fibrillation, and less common arrhythmias such as Brugada syndrome. A significant number of associated variants have been found to systematically localize in non-coding regulatory elements that control the tissue-specific and temporal transcription of genes encoding transcription factors, ion channels, and other proteins. However, the identification of causal variants and the mechanism underlying their impact on phenotype has proven difficult due to the complex tissue-specific, time-resolved, condition-dependent, and combinatorial function of regulatory elements, as well as their modest conservation across different model species. In this review, we discuss research efforts aimed at identifying and characterizing-trait-associated variant regulatory elements and the molecular mechanisms underlying their impact on heart rate or rhythm.

## 1. Introduction

Cardiac arrhythmias represent a highly prevalent clinical problem. Cardiac arrhythmias can present as a stand-alone problem or as a complication of underlying heart disease. As a result, the etiology of arrhythmias is highly variable, ranging from electrophysiological causes (e.g., ion channelopathies and/or medication-induced abnormalities), to previous ischemia/infarction, stress, autoimmune diseases, and aging [1,2,3,4,5]. In the past, arrhythmias were considered to be largely acquired, resulting from environmental factors or other diseases. However, human genetic studies identified sporadic familial arrhythmias caused by rare variants (mutations), typically in coding exonic sequences, that have a large effect size and show near Mendelian heritability [6]. Such rare variants are often found in genes involved in membrane depolarization or repolarization of the cardiomyocytes, or action potential (AP) impulse propagation. Examples of such genes with arrhythmia-causing coding variants include the cardiac sodium channel encoding *SCN5A* (Brugada syndrome, Long QT syndrome type 3) and potassium channel encoding *KCNQ1* (Long QT syndrome type 1) [7]. Their biological effects can be explained by loss or change of function of the affected protein due to an early stop codon or a changed or inserted amino acid, as seen, for example, in *SCN5A* associated with the Brugada syndrome [8]. Further, some of these variants were found to create or disturb splice sites, thereby altering protein sequences in a non-canonical way [9].

Since 2006, genome-wide association studies (GWAS) have identified many common genetic variants (SNPs) associated with electrocardiographic and heart rate variability traits (pacemaking, autonomic function, conduction, repolarization) [10,11,12,13,14,15,16,17]. This has led to the recognition of a large degree of genetic susceptibility to cardiac arrhythmia. This genetic component is commonly referred to as genetic predisposition and is composed of many rare and common genetic variants such as single nucleotide polymorphisms (SNPs), of which each individual has millions across their genome [18,19]. The common variants are often located in non-coding regions of the genome, which represents 98% of the total genome, and have low effect sizes that incrementally increase the risk of developing a particular arrhythmia such as atrial fibrillation [10]. Non-coding variants influencing traits or disease risk have been observed to systematically map to and potentially influence the function of regulatory DNA elements (REs) that control spatial and temporal transcription. In this review, we will focus on genetic variants in non-coding regulatory DNA elements (REs) that have been associated with rhythm and conduction traits or arrhythmia, the mechanisms underlying their impact on the regulation of gene expression, and downstream effects on rate or rhythm. We will limit the review to variant-containing regulatory elements (REs) for which experimental evidence regarding their impact on heart rate, conduction, or rhythm is available (See Table 1).

## 2. Heart Rate and Rhythm, ECG, and Genetic Association

Heart rate and rhythm describe the spatio-temporal pattern of action potential (AP) generation and propagation through the heart. We briefly introduce basic concepts related to generation and conduction of an AP, how this translates to ECG parameters, and what the ECG parameters used in GWAS studies mean. For detailed descriptions of cardiac electrophysiological processes and insights into mechanisms of normal and abnormal rate and rhythm, we like to refer to excellent reviews on cellular electrophysiology [45], impulse propagation [46], and ion channels [47].

In each individual cardiomyocyte, an AP is triggered by depolarization of the membrane through gap junctional currents from adjacent myocytes (Figure 1A, top right), cytosolic calcium ion concentrations increase, calcium is released by the sarcoplasmic reticulum, and contraction ensues in a process called excitation-contraction coupling [48]. The ventricular AP has five major phases, as diagrammed in Figure 1A (bottom right). First, a rapidly depolarizing inward sodium current is induced (phase 0), which is dominantly driven by the cardiac sodium channel Na_V_1.5 (encoded by *SCN5A* in mammals). A transient outward potassium current generates the short-lasting repolarization of phase 1. During phase 2 (plateau phase), a depolarized state is temporally maintained by a balanced combination of inward calcium and outward potassium currents. During phase 3 (repolarization phase), the inward calcium current diminishes and the outward potassium currents rise, repolarizing the membrane potential to its resting level and readying the cell for a new cycle of excitation and contraction. During phase 4, the cell is considered at rest, and the depolarized state is maintained by, amongst others, inward rectifier potassium channels. Pacemaker cardiomyocytes of the SAN (and to a lesser extent of the AV node and ventricular conduction system) show spontaneous depolarization during phase 4 of the AP (Figure 1A, bottom left). This spontaneous activity, which is possible due to the absence of inward rectifier potassium channels in SAN cells, is driven by a combination of the so-called ‘voltage clock’, referring to the voltage-driven pacemaker (or ‘funny’) current I_f_ (flowing through HCN channels during the ‘diastolic depolarization’ of phase 4), and the so-called ‘calcium clock’, referring to the subsarcolemmal Ca^2+^ releases that occur during the diastolic depolarization and activate the sarcolemmal sodium-calcium exchange current [49,50,51]. This spontaneous activity is further modulated by neurohumoral input [52].

As the AP in cardiomyocytes triggers an AP in adjacent cardiomyocytes, a depolarizing wave is set in motion that rapidly propagates through the cardiac muscle. The velocity of conduction of the AP greatly depends on the duration of phase 0. SAN pacemaker cells and atrioventricular node cardiomyocytes have a relatively long phase 0 (slow ’upstroke’) and propagate APs slowly. Atrial and ventricular cardiomyocytes have a short phase 0 (fast ‘upstroke’) and propagate rapidly, with the specialized ventricular conduction system cardiomyocytes propagating the AP and activating the ventricles very rapidly. The expression levels of *SCN5A* as well as of low- and high-conductance gap junction channels (i.e., *GJC1* (Cx45), *GJA1* (Cx43), and *GJA5* (Cx40), respectively) in these different cells correlate well with the conduction velocities in the components they form. For example, expression levels of *SCN5A* are very low in the SAN and atrioventricular node, high in the atria and ventricles, and very high in the ventricular conduction system [53].

The electrical activation and deactivation of the heart has an intrinsic spatio-temporal pattern that can be visualized by an ECG, showing voltage versus time, and that is directly related to the contraction pattern of the heart through the aforementioned excitation-contraction coupling (Figure 1B). Pacemaker cells in the sinoatrial node spontaneously generate APs, which propagate to the surrounding atrial working cardiomyocytes. This, in turn, causes en masse depolarization of the atrial myocardium, which is visible on an ECG as the P-wave. The electrical signal is propagated to the ventricular conduction system via the atrioventricular node, in which impulse propagation is delayed to allow complete contraction of the atria and filling of the ventricles before ventricular activation occurs. This delay is reflected by the PR interval on an ECG. The electrical impulse is then rapidly conducted through the AV bundle and bundle branches in the ventricular septum, and subsequently through the Purkinje fibers to propagate and distribute the depolarizing impulse throughout the ventricular muscle. Synchronized depolarization of the ventricular cardiomyocytes is represented by the QRS complex on the ECG, whereas subsequent repolarization of the ventricles is reflected by the T-wave. Deviations from “normal” ECG morphology represent abnormalities in the patterns of depolarizing (and repolarizing) waves through the heart compartments, revealing abnormal heart rate or rhythm. These deviations from “normal” ECG morphology have been used as traits in GWAS.

GWAS are population genetic studies investigating the association of genetic variants in the human population with specific phenotypes (traits, diseases). These common variants typically possess low individual pathogenicity, as opposed to rare but high impact variants/mutations. Quantitative electrocardiographic traits investigated in GWAS include QT interval and QRS duration, PR interval, heart rate, heart rate variability, and heart rate response to exercise and recovery (see https://www.ebi.ac.uk/gwas for an overview (accessed on 25 September 2023)) [54]. While these traits do not represent rhythm disorders, they are biomarkers for increased risks of developing an arrhythmia, such as AV block and atrial fibrillation (AF) (PR interval) and sinus node dysfunction ((variability in) heart rate; HR), and more generally have been associated with morbidity and mortality [55,56]. In addition, GWAS have uncovered a large number of SNPs associated with AF, made possible by the large number of registered AF cases [22,57,58,59]. Genetic variation associated with much less prevalent arrhythmias, such as Brugada syndrome (BrS), has also been investigated, yielding very informative common variants at specific loci in the genome [40,60,61]. From these as well as other studies, three observations and their implications stand out: (1) the vast majority of SNPs associated with heart rate, rhythm, and conduction traits, and arrhythmia are found in non-coding DNA, and have been found to be enriched in putative transcriptional REs [62,63,64]; (2) the lead SNPs are in linkage disequilibrium with a large number of other SNPs, and it is not known which of these SNPs or combination thereof are causal; (3) the genes closest to the SNPs show overlap between the studies of different traits and arrhythmia, and are enriched for essential cardiac transcription factors and ion channels [11,56,65].

## 3. Regulatory Elements

REs are small (typically tens to hundreds of base-pairs) non-coding regions in the genome that regulate gene transcription. The function of these elements is crucial for normal development, cell and tissue homeostasis, responses to external stimuli, and ultimately survival of the organism [66]. REs typically contain motifs (specific patterns of nucleotides) that are recognized by transcription factors that complex with other nuclear proteins and RNAs involved in chromatin remodeling or conformation, such as co-activators, histone-modifying enzymes, etc., to regulate the transcription rate of their target genes. The human genome contains hundreds of thousands of putative REs, tens of thousands of which show activity in one or more cell types, with any one gene potentially being controlled by several REs [67]. These elements function in a cell-type- and epigenetic-state-specific manner as well as responding to extra-cellular signals, allowing dynamic and precisely tuned gene expression matching requirements for cellular differentiation and cell and organ function.

REs can regulate gene transcription in several ways. REs directly surrounding a transcription start site (TSS), classically called promoters, directly initiate and control rate of RNA transcription by binding and releasing components of the pre-initiation complex near the TSS. REs more distally located to a transcription start site, classically called enhancers or repressors, typically bind transcription factors, which in turn recruit proteins and ncRNA complexes to these sites and interact with promoters and/or the pre-initiation complex to stimulate or suppress transcription initiation or elongation [68]. These proximal and distal REs are organized in dynamic 3D structures (loops, topologically associated domains (TADs), compartments), which allow interaction between sequences that are far apart when plotted on the linear DNA [69]. These 3D structures play another important role in regulation, which will be discussed later on. These RE-bound protein-RNA complexes also recruit chromatin-modifying enzymes, which alter histone side chains (e.g., acetylation, methylation) [70]. Histone modifications can be recognized by reader proteins that in turn influence gene transcription, influencing the chromatin structure itself; accessible and transcriptionally competent vs. tightly packed, inaccessible, and transcriptionally inactive (collectively called epigenetic state). Thus, the REs play a defining role in gene regulation and cell state [67].

While a subset of REs is evolutionarily conserved, many show evolutionary divergence [71,72], and several studies have suggested that RE evolution is an important driver of speciation, i.e., the difference in regulatory landscapes underlies differences between species [66,73]. Moreover, the inter-individual variations in the sequence of REs are also likely to importantly contribute to differences between individuals from the same species, including differences in traits and susceptibility to particular diseases [74,75,76]. Single genetic variants frequently affect binding of transcription factors, affecting local chromatin state and transcription, implicating these natural genetic variants in phenotypic heterogeneity [77,78,79].

Variation in REs that perturb their function typically leads to a milder or more tissue-restricted phenotype when compared to a pathogenic coding variant in the REs target gene. The coding variant will affect the function of the protein in every tissue where the gene is expressed. REs, on the other hand, act tissue- or cell-state-specifically, and therefore variants in REs affect the expression level of their target gene(s) in particular tissues, whereas other tissues remain unaffected (Figure 2) [80]. Moreover, since REs are usually composed of several binding sites, and often in complexes of several REs that together regulate transcription of a target gene, some of which may be partially redundant, variation in a single transcription factor binding site is unlikely to disrupt complete RE function [81].

The genome is organized in highly dynamic loops and TADs [82] that cluster together in A and B compartments, which roughly correspond to active euchromatin and inactive heterochromatin, respectively [83]. Organization of the genome in TADs has been suggested to allow for interaction between REs and their target genes, thereby allowing REs to exert their function, while preventing interaction between REs and genes in other TADs, thus imposing specificity. However, it should be noted that TADs do not seem to solely control whether REs can impact target gene expression [84,85]. Chromatin looping and TAD organization is mediated by several proteins, most prominently CTCF and Cohesins. TADs are suggested to form by loop extrusion of chromatin through Cohesin complexes until a boundary is encountered, for example convergent-oriented CTCF-bound elements [69,86,87]. Structural variation as well as mutations in CTCF motifs can cause reorganizations in the topology of the genome, and “fusion” of TADs, leading to gene expression changes and possibly disease [88,89,90]. Indeed, common variants can modulate CTCF binding sites and long-range chromatin contacts, implicating these variants in chromosomal architecture [79].

While a large fraction of common variants influencing rhythm or conduction traits or arrhythmia risk are thought to influence RE function, for only a few has strong functional evidence been provided. Transcription factors that control gene regulation underlying heart development, acquisition and maintenance of cell identity, and homeostasis do so in a highly dose-dependent manner [91]. Their misexpression at any stage of development in the mature heart may lead to arrhythmia predisposition [11,92,93]. Therefore, REs that control expression of genes encoding transcription factors are expected to be very sensitive to genetic variation that influences their activity (Figure 3). This may explain why loci harboring genes encoding such transcription factors display recurring hits in GWAS for cardiac traits. For several of these transcription factor gene loci, including *PRRX1, TBX5, TBX3, ZFHX3, PITX2,* and *HAND1*, REs that regulate their expression and are influenced by trait associated variation have been identified (Table 1). In addition, variant REs have been identified for genes encoding proteins that importantly contribute to heart rate, rhythm, or conduction properties such as ion-channel-encoding genes *HCN4*, *SCN5A,* and *SCN10A* (Figure 3). We will discuss several examples of REs containing common genetic variants causally related to cardiac electrophysiological phenotypes.

## 4. Variant REs for Transcription Factor Genes Implicated in Arrhythmia and ECG Trait GWAS

### 4.1. PRRX1

*PRRX1* encodes paired related homeobox 1 that is expressed most prominently in mesenchymal cells and important for their differentiation and development. The *PRRX1* locus contains multiple common variants associated with AF [58]. Moreover, eQTL analysis indicated that lower *PRRX1* expression in atrial tissue is associated with AF [22,57]. An RE was identified in the upstream non-coding region that was able to drive reporter expression in zebra fish heart and skeletal muscle and cultured HL-1 cells [33]. Common variant rs577676 affected the activity of the RE in HL-1 cells and transgenic zebra fish, with the risk allele reducing RE activity.

To explore the function of the variant non-coding region including the RE in vivo, the orthologous region was deleted from the mouse genome [34]. *PRRX1* is expressed at low levels in cardiomyocytes and at relatively high levels in non-cardiomyocytes (mesenchymal cells) in the heart. Mice with the deletion of the RE-containing region lost *Prrx1* expression specifically from cardiomyocytes, whereas the cardiac expression of other genes positioned close to the deletion and normally expressed in the heart was not affected. These findings indicated that the deleted region harbors an RE that specifically controls transcription of *Prrx1* and only in cardiomyocytes. A large set of genes was deregulated in cardiomyocytes of these mice, and further analysis of the changes in chromatin accessibility suggested that PRRX1 antagonizes MEF2 function at many target REs. These mice showed altered calcium handling and atrial arrhythmia sensitivity. Together, these studies suggest that the activity of a cardiomyocyte-specific RE driving *PRRX1* expression could perceivably be perturbed by a common variant associated with increased AF risk, leading to reduced expression of *PRRX1* in cardiomyocytes of risk allele carriers and deregulation of atrial gene expression and function.

### 4.2. TBX5

The *TBX5* locus has been a recurring hit in GWAS, and several variants in non-coding regions in and around *TBX5* were found to associate with AF and ECG traits (see https://www.ebi.ac.uk/gwas for an overview (accessed on 25 September 2023)) [54]. *TBX5* is transcription factor which is involved in the development of the limbs and heart, and mutations in *TBX5* have been shown to cause Holt–Oram syndrome [94]. A landmark study in 2012 showed that a mutation in a *TBX5* RE was associated with a case of isolated congenital heart disease, atypical for *TBX5* loss-of-function mutations and Holt–Oram syndrome [95]. This study supported the hypothesis that variation in REs leads to milder and partial phenotypes, compared to mutations in the coding region of a gene.

*TBX5* is crucial for correct cardiac and conduction system development and homeostasis and maintenance of atrial identity [23,94,96,97,98,99]. *TBX5* is surrounded by a large non-coding region (gene desert) and shows a complex expression pattern, indicative of a complex regulatory landscape with many REs. In the developing mouse heart and human cardiomyocytes, *TBX5* resides in a separate TAD, and the interaction between adjacent TADs is limited [100], indicating that the variants in REs in the *TBX5* TAD may influence *TBX5* expression, whilst not affecting expression of genes in adjacent TADs, such as *TBX3*. The *TBX5* promoter was observed to interact with a region in the last intron of *TBX5*, which also showed cardiomyocyte ATAC-seq (chromatin accessibility) and H3K27ac signals in human cardiomyocytes, typical of active REs [23]. This human *TBX5* intronic region also forms part of the larger AF-associated region containing several AF-associated variants, and is evolutionarily conserved. Deletion of the mouse orthologous intronic region caused very modest increase (30% more mRNA) in the expression of *Tbx5* in postnatal atria of mice. Cardiac expression of other genes in the vicinity—*Tbx3*, *Rbm19,* and *Med13l*—was not affected, underlining the selectivity of the RE for *Tbx5,* both present within the same TAD. Moreover, mice carrying the intronic RE deletion showed increased RR intervals, heart rate variability, PR intervals, sinus node recovery times, and Wenckebach cycle lengths. Additionally, increased atrial arrhythmia inducibility and duration were seen. Cellular electrophysiology studies showed increased action potential durations. This observation—that a very modest increase (approximately 1.3-fold increase in mRNA) in atrial *Tbx5* expression results in arrhythmia susceptibility—aligns with the observation that AF risk in humans has been associated with slightly increased, not decreased, expression of *TBX5* in cardiac tissues [22,57]. Moreover, it shows that even a small increase in expression of a transcription factor gene can have physiologically relevant consequences, such as increased disease predisposition (Figure 3).

Risk allele of SNP rs7312625 in the intronic AF-associated region significantly increased RE activity in luciferase assays in HL-1 atrial cardiomyocyte-like cells, indicating it could be causally related to increased *TBX5* expression and AF predisposition. However, the mechanism of action of this SNP is not completely clear yet. SNP rs7312625 disrupts a predicted binding site for LIN54, which is a component of the DREAM complex [101]. The DREAM complex has been associated with cell-cycle-dependent repression of genes [102]. Perhaps relevant in this context, reduced expression of *LIN54* in human atrial tissues was recently associated with AF predisposition [57].

### 4.3. TBX3

*TBX3* encodes T-box factor 3, a member of the T-box transcription factors that is required for the development of mammary glands, limbs, and lungs. *TBX3* function is dose-sensitive; heterozygous loss-of-function mutations in this gene cause Ulnar Mammary syndrome, and it is frequently found to be ectopically expressed in a wide range of epithelial- and mesenchymal-derived cancers [103]. *TBX3* is critical for correct development of the cardiac conduction system, including sinus node, atrioventricular node, and bundle [96]. Additionally, several studies have shown that both pre- and postnatal loss of *Tbx3* results in arrhythmia predisposition [104], while signaling upstream *Tbx3* leading to reduced *Tbx3* expression leads to loss of AV-junctional cell identity [105].

*TBX3* is located in a gene desert of more than 1Mb, flanked by *TBX5* (~260 kb downstream) and *MED13L* (>1 Mb upstream). Several GWAS have found associations between regions in this gene desert and PR interval, heart rate recovery after exercise (HRRAE), and ECG (see https://www.ebi.ac.uk/gwas for an overview (accessed on 25 September 2023)) [54].

Two distinct regions containing rhythm-associated variation were investigated for RE potential, including a region 85–6 kb upstream *TBX3* [25]. This region contains SNPs associated with PR interval [24,27]. The orthologous mouse region harbors Res controlling *Tbx3* expression in the atrioventricular conduction system [100]. Deletion of a 51 kbp mouse orthologous region (called VR2) caused an increase in expression of *Tbx3* in the sinus node and atrioventricular conduction system. Expression of nearby genes *Tbx5*, *Med13l,* and *Rbm19* was not affected, indicating the potential REs in this region selectively regulate *Tbx3*. Additionally, expression of target genes *Hcn4* and *Cacna1g* was increased, while *Scn5a* expression was reduced. These mice showed reduced PR interval and increased QRS duration. Further dissection of this region revealed several potential REs, which also showed variable activity between reference and risk allele in a non-cardiomyocyte and cardiomyocyte cell line. However, a precise mechanistic relation between variants in the REs and the regulation of *Tbx3* was not found [25].

*TBX3* neighboring gene *MED13L* has also been a recurring hit in GWAS. A region ~170 kb downstream of *MED13L* contains SNPs associated with PR interval and HRRAE [24,26,106]. The GWAS assigned the variant region to the closest gene, *MED13L*, as well as several unknown transcripts from this gen–e desert. However, even though the HRRAE-associated variant region is positioned about 1 Mb upstream of *TBX3*, it resides in the TAD with *TBX3*, whereas *MED13L* is located in an adjacent TAD, with a CTCF-rich boundary in between the two TADs. In fact, Hi-C and 4C-seq data showed the HRRAE region is in close vicinity to *TBX3* in 3D space [28,100]. This HRRAE-associated region was shown to contain several human pacemaker-cell-specific REs (RE1-RE2) that could drive reporter expression in the developing sinoatrial node of mouse [28]. Interestingly, deletion of the mouse orthologue of the entire region (~280 kb) harboring the HRRAE-associated variants and the pacemaker-cell-specific REs, abolished *Tbx3* expression specifically in the sinoatrial node pacemaker cells and innervating neurons, and not in other parts of the conduction system or any other *Tbx3*-expressing tissue. Furthermore, the deletion did not affect *Med13L* or *Tbx5* expression levels. These mice showed increased heart rate variability and increased sinus node recovery times. It was also found that an SNP in LD with HRRAE lead SNP rs61928421, rs140828160, was located in RE2. When the minor allele of this SNP was introduced, the enhancer activity of RE1-RE2 was reduced in transgenic mouse embryos [28]. These data indicate a mechanism in which HRRAE variants reduce the activity of pacemaker-cell-specific REs for *TBX3*, leading to reduced expression of *TBX3* in the pacemaker cells and innervating nerves, affecting HR and HRRAE.

### 4.4. ZFHX3

*ZFHX3* encodes Zinc Finger Homeobox 3 (AT Motif-Binding Factor 1) and has been implicated in promotion and suppression of cancers, in regulating neuronal differentiation, and in regulating circadian function in the Suprachiasmatic Nucleus [107]. The locus harboring *ZFHX3* is associated with AF [22,57,58]. Recently, loss-of-function experiments in mice revealed that *Zfhx3* regulates a large number of atrial genes and signaling pathways required to maintain normal atrial function [35]. Loss of *Zfhx3* resulted in atrial dilatation and arrhythmia. Which of the associated variants causes the AF risk remained elusive. However, rs12931021, in an intron of *ZFHX3*, was observed to influence activity of a putative RE and influenced the epigenetic state of that RE in hiPSC-cardiomyocytes [35]. Moreover, when the RE was deleted in hiPSC-cardiomyocytes, *ZFHX3* expression was reduced. This was confirmed in hiPSC-cardiomyocyte lines homozygous for the risk (AA) or protective (CC) allele, in which the risk allele lines expressed *ZFHX3* at lower levels. These data suggest that the AF-associated variant reduces RE activity and *ZFHX3* expression in risk-allele carriers, contributing to the likelihood of developing AF.

### 4.5. PITX2

Of all variants associated with AF, those at chromosome 4q25 have been most strongly associated, reaching extraordinary significance (*p* < 10 exp-710) in a recent GWAS [57]. In fact, multiple independent association signals were identified at this locus, clustering in a non-coding region between about 50 and 200 kbp upstream of *PITX2* [108,109]. Despite the strong association, the mechanism linking variation to AF predisposition has remained poorly understood. *PITX2* encodes the paired-like homeodomain transcription factor 2 with a well-established role in left-right patterning and cardiogenesis, and is expressed in the left-sided heart structures, including the left atrium and pulmonary vein, but its expression is not detected in the right atrium This asymmetric expression has been found to be involved in specification of the primary pacemaker to the right sino-atrial junction, with disruption resulting in right atrial isomerism [110,111,112]. Left atrium and pulmonary vein expression of PITX2c is maintained in the adult mammalian (human) heart [11,113]. Heterozygous loss-of-function mouse models and human iPSC loss-of-function models indicate that PITX2 (the *PITX2c* isoform transcribed from a distinct promoter) is also involved in regulating the electrophysiological properties of cardiomyocytes [114,115,116]. These studies point to several different possible mechanisms by which aberrant *PITX2* expression may predispose to arrhythmogenesis [117]. Therefore, it is generally assumed that the AF-associated variants in the non-coding region influence the function of REs for *PITX2*, presumably leading to lower expression levels of *PITX2* in the left atrium and/or pulmonary vein myocardium. Loss-of-function mutations in *PITX2* cause Axenfeld–Rieger syndrome, a congenital defect syndrome affecting development of teeth, eyes, and the abdominal region [118]. Arrhythmogenesis has not (yet) been well investigated in patients with this syndrome.

Scanning a large genomic region including the AF-associated region for RE activity in HL1 cells and transgenic embryos, a putative non-cell-type-specific RE was identified. Using 3C, the region harboring the RE was found to interact with *Pitx2* and *Enpep*, suggesting it may regulate both these genes [119].

One of the independent AF-associated regions overlaps the *PITX2* gene itself. To identify regulatory variants among the associated variants in LD with the lead SNP (rs1448818) [29] in this 84 kbp region, the entire region was scanned for the presence of putative REs [30]. Based on epigenetic signatures and activity in zebrafish reporter assays, six putative REs were identified, distributed across the 84 kbp genomic segment. Next, they tested small DNA regions surrounding each SNP in the six REs in luciferase assays to compare RE activity between the two alleles in HL-1 atrial cardiomyocyte-like cells. Two SNPs were found to reduce RE activity, one of which (rs2595104) in an intron of *PITX2a/b* (and upstream of the promoter of *PITX2c*) was also associated with reduced *PITX2c* expression in human-stem-cell-derived cardiomyocytes. This differential activity was mediated by transcription factor TFAP2a, which bound robustly to the non-risk allele but not to the risk allele. This study provided the first evidence for a causal relation between risk variant and lower expression of *PITX2c* in cardiomyocytes [30].

The prevailing hypothesis that variants at 4q25 in the large region upstream of *PITX2* affect *PITX2* expression has been tested in mice, in which evolutionarily conserved REs were identified in the genomic region orthologous to the human AF-associated variant region [31]. Deletion of one such putative RE region caused downregulation of atrial *Pitx2* expression and AF susceptibility specifically in male mice. Chromatin conformation capture analysis revealed the RE deletion altered the *Pitx2* gene body to a transcriptionally less active chromatin state. Other genes in the vicinity of the deletion (e.g., *Enpep* and *Sec24*) were not affected. These data indicate that variants at 4q25 may exert their effect via altering the activity of REs that influence *PITX2* expression and increase arrhythmia predisposition in a cell-state- and sex-dependent manner.

### 4.6. HAND1

HAND1 is a transcription factor involved in development of the heart, and mutations in its gene have been implicated in congenital heart defects [120,121]. Additionally, knockout mice models show that loss of *HAND1* expression during development results in cardiac conduction system defects leading to arrhythmogenic conditions [122,123]. Interestingly, postnatal loss of *HAND1* expression does not seem to impact cardiac conduction, while postnatal overexpression does result in a predisposition to arrhythmia in mice [124,125].

GWAS studies have found associations between the *HAND1* locus and QRS duration and QT interval [27]. Many associated SNPs concentrate in an area 15–7 kb upstream of *HAND1*, which coincides with a highly conserved genomic region. This conservation was used to identify a GATA-dependent RE, Hand1LV, in the mouse genome, which was shown to be necessary and sufficient to drive left ventricular *Hand1* expression in transgenic mice [21]. Mice in which the 750 bp RE was deleted showed a hyperplastic ventricular conduction system, decreased PR intervals, and increased QRS durations based on abnormal ventricular activation. Two SNPs were identified in human HAND1LV in LD with QRS duration-associated SNP rs13165478, and showed that one, rs10054375, impeded GATA4 binding to HAND1LV. Because the sequence at the location of the SNPs is conserved between mouse and human, mice carrying the minor allele of both SNPs in Hand1LV were generated. Mice homozygous for the minor allele SNPs were found to have significantly reduced *Hand1* expression in the fetal left ventricle, although not fully resembling the reduction observed in Hand1LV deletion mice. These mice showed no abnormal phenotype on ECG, nor a hyperplastic conduction system, but did show aberrant activation maps as expressed by abnormal ventricular breakthrough patterns [21]. Together, this study indicates that QRS duration and QT-interval-associated SNPs in the *HAND1* locus may reduce the activity of RE HAND1LV, causing left-ventricle-specific reduction in *HAND1* expression during development (and in the adult), affecting morphogenesis and function of the conduction system.

## 5. Variant REs for Ion Channel Genes Implicated in Arrhythmia and ECG Trait GWAS

### 5.1. SCN5A and SCN10A

The *SCN5A* locus has often been implicated in cardiac arrhythmias [126]. *SCN5A* encodes the cardiac sodium channel Na_V_1.5, which is responsible for the fast inward sodium current (I_Na)_, which results in rapid depolarization in phase 0 of the action potential, typical for working myocardium as well as ventricular conduction system cardiomyocytes (Figure 1). Mutations in the coding region of *SCN5A* have been implicated in several arrhythmias, among others Brugada Syndrome, AF, and sick sinus syndrome [127]. Arrhythmias have been associated with both reduced expression of *SCN5A*, leading to conduction slowing, as well as gain-of-function mutations, leading to increased late sodium entry and other pathological changes (Figure 3) [128]. *SCN5A* forms a gene cluster with two other sodium channels *SCN10A* and *SCN11A*. Common variants associated with ECG parameters, AF, and Brugada syndrome have been identified across the *SCN5A*-*SCN10A* locus [40,57,129,130]. A large panel of putative REs across this region harboring QT interval-associated variants was tested in vitro, yielding seven variant REs, including one in *SCN10A*, that potentially impact on *SCN5A* expression [131]. However, their role in *SCN5A* regulation in vivo remains to be validated. The *SCN5A-SCN10A* locus harbors a number of REs that drive heart-specific expression of *SCN5A,* being evidenced by chromosome conformation capture assays, in vivo RE activity assays, and mice carrying deletions of the orthologous REs [37,38,132,133]. Downstream of *SCN5A*, a multi-RE (RE6-9), dubbed a super enhancer, has been identified and functionally validated in vivo [37]. Two of the REs of this enhancer harbor common variants associated with QRS interval (rs6810361 and rs6781009), both shown to reduce the activity of the respective REs [37,38]. Although it is tempting to assume that these variants will affect *SCN5A* expression in the human heart, formal proof has yet to be provided.

Many GWAS involving *SCN5A* typically also find consistent associations between variants in and around the neighboring gene *SCN10A* and conduction parameters [20,39] and Brugada syndrome [40,60]. *SCN10A* encodes Na_v_1.8, a voltage-gated sodium channel expressed in the nervous system and involved in pain and tactile sensation [134]. Prior to the GWAS linking *SCN10A* to cardiac arrhythmias, no cardiac function of *SCN10A* had been described, and expression of *SCN10A* in the heart had not been detected. The new GWAS signal, supplemented with expression data, showed expression of *SCN10A* in isolated cardiomyocytes [39], and enriched *SCN10A* expression in the ventricular components of the CCS [20].

Using ChIP-seq data for cardiac transcription factors, it was shown that a GWAS lead SNP associated with QRS interval, rs6801957, localizes in and disrupts a TBX3/5 transcription factor binding site in a putative RE in an intron of *SCN10A* (RE1-2) [132]. Both mouse and orthologous human putative regulatory fragments were shown to have enhancer activity in the heart (interventricular septal region) of transient transgenic mouse embryos. Using a zebrafish reporter assay, it was shown that the minor allele of rs6801957 indeed reduces enhancer function in the heart. This study was later followed-up using 4C chromosome conformation capture in mice [133], providing evidence for an interaction between the intronic RE and the *SCN5A* promoter, while showing only nominal interaction with the *SCN10A* promoter. This provided support for an effect of the intronic SNPs on the activity of an RE for *SCN5A.* This hypothesis was strengthened by expression analyses, which found significant expression of *SCN5A*, but not *SCN10A*, in mouse and human hearts and that the minor allele of rs6801957 reduced activity of the RE and *Scn5a* expression in transgenic mice carrying large reporter constructs. Furthermore, a significant correlation was observed between SNP rs6801957 genotype and *SCN5A* expression in human heart samples, with homozygous carriers of the minor allele (AA) expressing less SCN5A than heterozygous (GA) or homozygous (GG) carriers. Together, these reports seemingly complete the picture on SNP rs6801957 by identifying it to be a causal variant, the mechanism of its effect (disruption of T-box binding site, reduced activity of RE), its target gene (*SCN5A*), and the associated direction of effect: reduced expression of *SCN5A* in particular regions of the heart, causing reduced sodium current and conduction slowing.

However, other data show that in human left ventricular tissue, SNP rs6800541 is in linkage disequilibrium with rs6801957, but is not an eQTL for *SCN5A* or *SCN10A* [135]. Later haplotype block analysis found that the haplotype containing rs6801957 does associate with reduced *SCN10A* expression, but does not have a significant effect on *SCN5A* [40,136]. Conflicting reports about the cardiac expression of *SCN10A* continued, with several reports finding expression of both mRNA and protein in human tissue [135,137] and protein and functional channel in intracardiac neurons [138]. Other reports could not detect Na_v_1.8 protein, full-length *SCN10A* transcript, or functional ion channels [139,140].

An interesting observation from the report of Gando et al. [139] was the failure to detect full-length *SCN10A* transcripts via RNA sequencing, though the study did note the presence of apparent reads in the last two exons. A subsequent report found expression of the last seven exons in sorted cardiomyocytes, cardiomyocytes of the atria, sinus node and ventricular conduction system, as well as in human right atrial and left ventricular tissue, albeit at very low levels, which they named *SCN10A-short* [41]. An intronic promoter was identified downstream of the above-mentioned intronic RE1-2 that supposedly drives *SCN10A-short* expression in mice and possibly humans. To further investigate the role of the *SCN10A* intronic RE1-2 and the mechanism underlying the effect of rs6801957 within the RE, mice were generated in which RE1 was deleted and in which a 3bp deletion was made in the T-box binding site also disrupted by rs6801957. In both lines, they observed reduced expression of *Scn10a-short* in the heart, whereas *Scn5a* expression was hardly affected. Moreover, rs6801957 (tagging the entire haplotype with all ECG-trait- and Brugada-syndrome-associated SNPs in LD) showed a cis-eQTL effect on *SCN10A* expression, not *SCN5A* expression, in atrial and ventricular tissue, consistent with the findings in the other two reports. Electrophysiological measurements of the mouse revealed atrial conduction slowing, atrial arrhythmia inducibility, SAN exit block, increased sinus node recovery times, and increased QRS duration. Additionally, isolated cells from the RE1- and 3 bp deletion showed reduced current density. Further analyses in an overexpression system revealed that the protein Na_v_1.8-short itself (encoded by *SCN10A-short*) does not function as ion channel. However, co-expression with Na_v_1.5 (encoded by *SCN5A*) showed significantly increased current density, while other channel kinetics were unchanged. These results indicate that the mechanism of pathogenicity of rs6801957 is indeed disruption of function of the intronic RE1, reduced expression of *SCN10A-short* in rs6801957 haplotype carriers, possibly causing reduced current density of Nav1.5, and thereby compromising cardiac impulse propagation (expressed by an increased PR interval and prolonged QRS duration), as seen in GWAS studies.

### 5.2. HCN4

*HCN4* encodes hyperpolarization-activated cyclic nucleotide-gated channel 4, which is one of the main drivers of the Na^+^/K^+^ pacemaker current (I_f_) in the sinoatrial node. HCN4 contributes to the diastolic depolarization seen in pacemaker cardiomyocytes and involved in generation and control of cardiac rhythm [141,142] (Figure 1). The HCN4 locus was found to contain multiple variants associated with AF and resting HR [22,57,58,143]. *HCN4* shares its topologically associated domain (TAD) with several genes in close proximity, and many intra-TAD interactions are observed. This includes interactions between the genomic region containing AF-associated SNPs and multiple genes within the TAD, including *HCN4*. This makes assignment of variant REs to specific target genes increasingly complex. Using an elegant approach of self-transcribing active regulatory region (STARR) sequencing in immortalized rat atrial cardiomyocytes, a 200 kbp genomic segment including the AF-associated region was functionally screened for REs active in atria [42]. STARR-seq revealed several REs (enhancers and repressors) overlapping AF-associated SNPs in this region as well as active RE-associated epigenetic marks. In a second screen, two variants in REs in this AF-associated region (rs6495063 and rs6495062) showed allele-specific RE activity. Subsequently, a 22.3 kbp region orthologous to the variant RE-containing region in upstream of *HCN4* was deleted from the mouse genome. Homozygosity for this deletion proved lethal around embryonic day 11.5, with *Hcn4* expression displaying a strong reduction in the forming heart at embryonic day 9.5. This phenotype is reminiscent of *Hcn4*-deficient mice [144]. A significant reduction in sinus node *Hcn4* expression was also seen in adult mice heterozygous for the deletion. As a sign of the complexity of this locus, significant changes in expression during cardiac development and in the adult sinus node were also found for *Neo1*, *Nptn,* and *Loxl1*, which share the TAD with *Hcn4*. Interestingly, these mice also displayed increased heart rate variability, increased PR intervals, and increased sinoatrial node recovery times in addition to AF inducibility. These functional readouts indicate that variation at this regulatory locus can impact electrophysiology, and thereby AF susceptibility [42]. However, it has proven difficult to dissect the contribution of the different genes differentially regulated in this model. *Hcn4* has a known and highly important role in the pacemaker current of the cardiac conduction system, and is therefore a likely candidate gene impacting the phenotype. However, it is interesting to note that the transmembrane protein *Nptn* has been found to have an expression pattern similar to that of *Hcn4*, with strong enrichment in the sinus node, AV node and bundle, and Purkinje fibers [145]. However, the function of *NPTN* in the CCS is unknown, and further research is needed to resolve the contribution of different genes affected. Additionally, although the presumptive target genes of the variant REs in the human locus can be inferred from this mouse model, the variant or variants causing the association and the molecular mechanism have not been identified yet.

## 6. Conclusions and Perspectives

Large-scale genetic testing in the form of GWAS has contributed significantly to the current insight that genetic predisposition contributes to arrhythmias risk. In this review, we have discussed several examples of well-studied arrhythmia GWAS loci for which some further mechanistic insight has been obtained. We have highlighted the various methodologies used, showing the significant effort needed to go from trait-associated variants to mechanistic understanding. Application of state-of-the-art technology, for example creation of single SNPs in the genome of human iPSC-cardiomyocytes, seems to be a powerful approach for identifying their biological effect [35], circumventing the requirement of mouse orthologues SNPs to exist [21,41]. While such methodologies will help in identification of causative SNPs and the effect size of individual SNPs and improve understanding of molecular mechanisms underlying arrhythmia risk, they are very laborious and limited to specific cell states outside their organ context. Moreover, many thousands of sequence variants have been linked to heart rate and rhythm traits and arrhythmia, many of which are expected to affect REs, highlighting the need for cell-state-specific, genome-wide, high-throughput functional assays to convert association into causation. Current efforts focus on the identification and functional interpretation of variants using single-cell-epigenomics [146]. By analyzing single-cell-epigenomic data from human tissues or organ model systems, these technologies and approaches enable the cell-type- and cell-state-specific mapping of REs harboring functional variants, and the impact of this variation on RE function. The next challenge will be to explore and validate the impact of identified variant REs on heart function.

Additionally to poor understanding of GWAS hits, a large part of total arrhythmia inheritance has not been accounted for at all. Most importantly, until recently, the combination of known risk alleles from GWAS into polygenic risk scores has changed little in clinical practice [10]. However, more recently successful attempts have been made to use polygenic risk scores as independent predictors of atrial fibrillation risk [147,148]. This shows the first clinical utility of ECG trait and arrhythmia GWAS, and future research with large GWAS for different arrhythmias may continue to contribute to clinical use based on polygenic risk scores. Accompanying fundamental research into mechanisms underlying the impact of common and rare variants on gene regulation, and the identification of affect genes, may one day lead to personalized druggable targets for high-impact variants.

## Figures and Tables

**Figure 1 cells-13-00004-f001:**
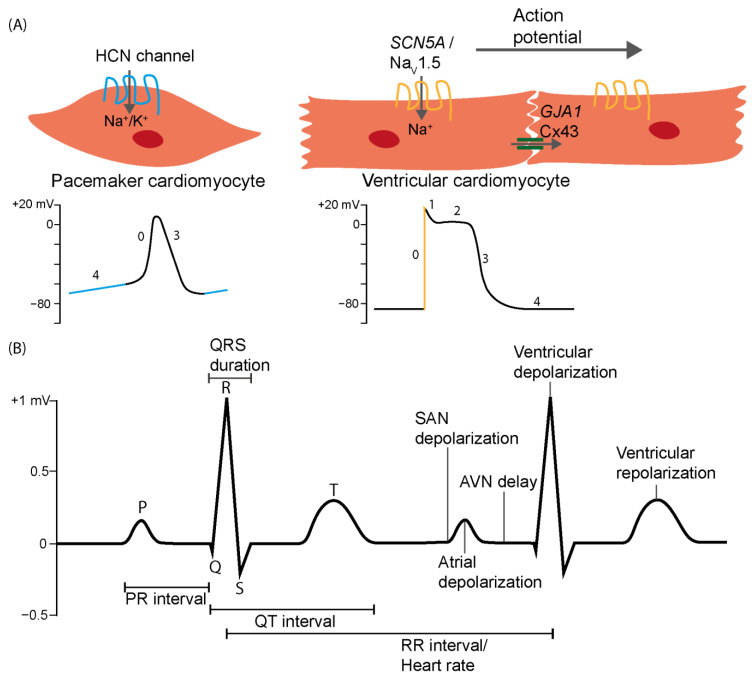
(**A**) Highly simplified depiction of pacemaker and ventricular cardiomyocytes, with their respective typical action potentials. Highlighted are the ‘funny’ current (phase 4 of action potential) mediated by HCN channels in pacemaker cardiomyocytes and the cardiac sodium current (active during phase 0) mediated by SCN5A in ventricular cardiomyocytes. (**B**) A typical ECG trace, showing the various activation and repolarization waves, as well as intervals. SAN, sinoatrial node; AVN, atrioventricular node.

**Figure 2 cells-13-00004-f002:**
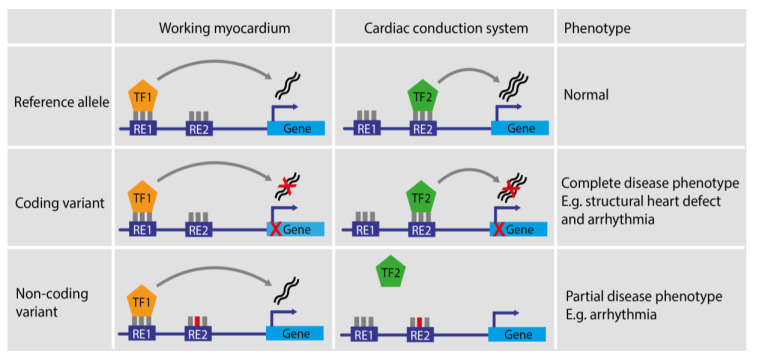
Effect of coding and non-coding variants in different tissues and their resulting phenotypes. In the reference allele situation, different Res control gene expression in different tissues. A coding variant leading to, for example, an early stop codon, will disrupt protein function in all tissues and time points where the protein is expressed, leading to compound disease phenotypes affecting multiple tissues. In contrast, a non-coding variant leading to disruption of RE2 will only affect tissues and time points in which expression of the gene is mediated by RE2, leading to partial disease phenotypes.

**Figure 3 cells-13-00004-f003:**
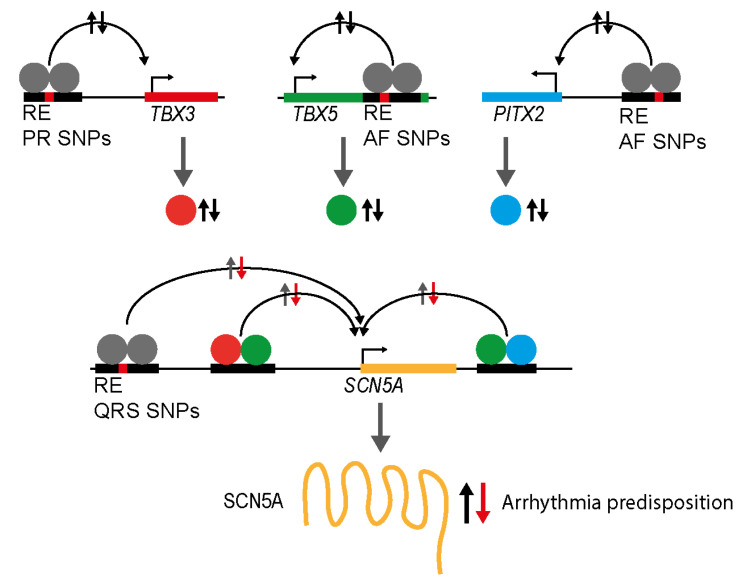
Overview showing mechanisms of SNPs in REs affecting expression of transcription factors, in turn changing levels and stoichiometry of transcription factors at REs of ion channel genes. These changes in transcription factor levels, as well as SNPs in REs controlling expression of ion channel genes, change the expression rate and protein level of, for example, SCN5A, leading to arrhythmia predisposition on reduced expression (red arrows).

**Table 1 cells-13-00004-t001:** Cardiac electrical phenotype GWAS implicated loci for which RE function has been suggested to cause functional variation.

GWAS Locus	GWAS Trait	Regulatory Element Name and Position, Related SNPs	Affected Genes	Evidence RE Function	Suggested Mechanism	References
*HAND1*	QRS duration in Europeans	Hand1^LV^ ~15kb upstream *Hand1* rs13165478 rs13185595 rs10054375	*Hand1*	Transgenic reporter assay of mouse fragment in mouse, mouse orthologue deletion	Risk allele reduces RE activity: Reduced LV specific HAND1 expression	[20] [21]
*TBX5*	Atrial fibrillation in mixed ancestry	TBX5^RE(int)^ Intronic of *Tbx5* rs7312625	*Tbx5*	In vitro reporter assay of human fragment in mouse atrial cell line, mouse orthologue deletion	Risk allele increased RE activity: Increased atrial TBX5 expression	[22] [23]
*TBX3*	PR interval, QRS duration in Europeans	VR2 ~85–6 kb upstream *Tbx3* rs11067264 rs6489973	*Tbx3*	In vitro reporter assay of human fragment in multiple cell lines, mouse orthologue deletion	Risk allele increases RE activity and decreases RE inducibility: Increased TBX3 expression in AVCS	[20] [24] [25]
*MED13L*	HRRAE in Europeans	RE1-RE2 ~1Mb upstream *Tbx3* rs61928421 rs140828160	*Tbx3*	Transgenic reporter assay of human fragment in mouse, mouse orthologue 280kb deletion	Risk allele reduces RE activity: Reduced TBX3 expression in SAN	[26] [27] [28]
*PITX2*	Atrial fibrillation in Europeans and Japanese	C Intronic of *Pitx2* rs1448818 rs2595104	*Pitx2*	Transgenic reporter assay of human fragment in zebrafish, EMSA, ChIp, human iPSC-CM deletion	Risk allele reduces RE activity: Reduced PITX2c expression in cardiomyocytes	[29] [30]
*PITX2*	Atrial fibrillation in mixed ancestry	E20 ~190 kb upstream *Pitx2* rs2129977 rs3853445 rs149829837	*Pitx2*	Mouse orthologue deletion	Risk allele reduces RE activity: Reduced PITX2c expression in left atrium in male only	[22] [31]
*PRRX1*	Atrial fibrillation in Europeans	E-F ~45 kb upstream *PRRX1* rs577676 rs3903239 rs10919449	*Prrx1*	Reporter assay of human fragment in zebrafish, mouse orthologue deletion	Risk allele reduces RE activity: Reduced PRRX1 expression in atrial cardiomyocytes	[32] [33] [34]
*ZFHX3*	Atrial fibrillation in mixed ancestry	- Intronic of *ZFHX3* rs2106261 rs12931021	*Zfhx3*	In vitro reporter assay of human fragment in human PSC-CM, ChIp, hPSC-CM deletion of SNP region	Risk allele reduces RE activity: Reduced ZFHX3 expression in atria	[22] [35]
*GJA1*	Atrial fibrillation in mixed ancestry	- ~680 kb upstream *Gja1* rs2816098 rs868155	*Gja1*	Mouse orthologue knockout	Risk allele reduces RE activity: Reduced GJA1 expression in atria	[22] [36]
*SCN5A*	PR interval, QRS duration In Europeans	RE6-9 ~3–25 kb downstream *Scn5a* rs6810361 rs6781009	*Scn5a* *(Scn10a)*	Transgenic reporter assay of human fragment in mouse, mouse orthologue deletion	Risk allele reduces RE activity: Reduced cardiac SCN5A expression	[24] [37] [38]
*SCN10A*	PR interval in Indian Asians, QRS duration in Europeans, Brugada syndrome in Europeans	RE1-2 Intronic of *Scn10a* rs6801957	*Scn10a* *(Scn5a)*	Transgenic reporter assay of human fragment in mouse, mouse orthologue deletion	Risk allele reduces RE activity: Reduced SCN10A-Short expression in atria/ventricular conduction system	[39] [38] [40] [41]
*HCN4*	Atrial fibrillation in mixed ancestry	- ~26–4 kb upstream *Hcn4* rs7172038 rs6495063 rs6495062	*Hcn4, Nptn, Neo, Loxl1*	In vitro MPRA of human fragment in human atrial cell line, mouse orthologue deletion	Risk allele reduces RE activity: Reduced expression HCN4 in SAN	[22] [42]
*CACNA1G*	PR interval, ECG morphology in Europeans	RE4-5 Intronic of *Cacna1g* rs757416 rs34081637	*Cacna1g, Epn3*	Mouse orthologue deletion	Changed expression level in AV node/conduction system	[43] [27] [44]

## Data Availability

Not applicable.

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
