# Peer review of "Role of Genetic Variation in Transcriptional Regulatory Elements in Heart Rhythm"

_cells, 2023, doi:10.3390/cells13010004_

Round 1
Reviewer 1 Report
Comments and Suggestions for Authors
This can be an insightful manuscript, yet there are some missing information:
1) The authors can add a few more relevant references related to GWAS: PMID 31321287
2) The authors need to include ARVC or ACM and desmosomal genes in their review (they can include some information from this article: PMID 29802319)
3) add some illustrations in section 2 about CCD
4) They need to add a section about cellular electrophysiology of the heart. Section 2 is still in the organ level, while section 3 is already in the molecular level. They need something to brigde those two sections. This article might be useful PMID: 32188566.
Comments on the Quality of English LanguageNo comment
Reviewer 2 Report
Comments and Suggestions for Authors
Jonker et al. reviewed the role of genetic variations in transcriptional regulatory elements in arrhythmias. This review summarizes well the knowledge to date regarding the role of REs located in the non-coding DNA regions in the common arrhythmias such as atrial fibrillation and inherited arrhythmias. Although GWAS identifies many SNPs in the non-coding DNA regions that are associated with disease, the underlying mechanisms are only partially understood, and this kind of review will be useful for understanding the current situation and for further development of future research.
Minor concern:
The description of line 85-130 cardiac conduction system is a bit redundant and could be summarized a little more.
Reviewer 3 Report
Comments and Suggestions for Authors
The review has summarized the reported genetic variants that are associated with heart rhythm. The review was generally well written, and the structure is clear. Here are some suggestions that can improve the manuscripts.
1. A few more example genes (e.g., KCNQ1) can be included in the section of ion channel proteins.
2. Aside from TF and Ion Channel proteins, would it be additional candidate genes from GWAS studies with other prominent functions to be discussed? Cataloging genes with other functions will make the review more comprehensive.
3. The table 2 is informative and it will be helpful if the authors can include the genetic backgrounds of each GWAS study.
4. When discussing the candidate genes (e.g. HAND1, HCN4, TBX3), some references might be missing at a few places when the findings were summarized. The audience might find it hard to refer to the original studies.
Reviewer 4 Report
Comments and Suggestions for Authors
The manuscript titled "Role of Genetic Variation in Transcriptional Regulatory Elements in Heart Rhythm," authored by Timo Jonker and colleagues, presents a comprehensive review of the current understanding of the role of genetic regulatory variation on ECG parameters and arrhythmias.
This manuscript offers a valuable and in-depth perspective on a topic of paramount importance to both biomedical researchers and cardiologists. It stands out for its clarity, and it is updated with the latest literature in the field of regulatory genetic elements in heart rhythm.
Furthermore, the manuscript effectively highlights the challenges associated with the identification and characterization of genetic variants within regulatory regions. It also elucidates the intricate task of elucidating their impact on cardiac electrophysiology and arrhythmias.
My only minor criticism is related to the figures included in the manuscript. While the first figure compares the impact of coding and non-coding variants, the second figure is an adaptation of a prior work by the research group. To enhance the manuscript's effectiveness, I recommend the inclusion of an additional figure that summarizes the key take-home messages derived from the review. Such an addition would serve to further strengthen the manuscript's impact and accessibility to readers.
Round 2
Reviewer 1 Report
Comments and Suggestions for Authors
Thanks for sending us the revised version of this manuscript. The addition of new section (section 2) is good but still very superficial, definitely could be improved. Perhaps, the authors could consult with basic EP scientists to improve this part.
Also, I don't see a nice coherence between section 2 and 3. The end of section 2 should be an opening to section 3.
Most of TBX families genes and HAND1 are found in growing heart, not in the adult human heart. Not sure what is the relevance in human cardiac arrhythmias?
What is the physiological role of PITX2?
Comments on the Quality of English LanguageNo comment
Round 3
Reviewer 1 Report
Comments and Suggestions for Authors
Thanks, this is indeed I would expect from such a review.